# Micromobility Users’ Behaviour and Perceived Risk during Meeting Manoeuvres

**DOI:** 10.3390/ijerph182312465

**Published:** 2021-11-26

**Authors:** Alejandra Sofía Fonseca-Cabrera, David Llopis-Castelló, Ana María Pérez-Zuriaga, Carlos Alonso-Troyano, Alfredo García

**Affiliations:** Highway Engineering Research Group, Universitat Politècnica de València, 46022 Valencia, Spain; alfoncab@cam.upv.es (A.S.F.-C.); anpezu@tra.upv.es (A.M.P.-Z.); caraltro@tra.upv.es (C.A.-T.); agarciag@tra.upv.es (A.G.)

**Keywords:** micromobility, bicycle track, instrumented e-scooter, meeting manoeuvre, clearance distance, perceived risk

## Abstract

Mobility patterns and lifestyles have changed in recent years in cities worldwide, thanks to the strong rise in modes of travel commonly referred to as micromobility. In this context, e-scooters have experienced a great rise globally which has led to an increase of crashes involving this type of micromobility vehicle in urban areas. Thus, there is a need to study e-scooter users’ behaviour and their interaction with cyclists. This research aimed at characterizing the meeting manoeuvre between micromobility users along diverse typologies of two-way bicycle track by using an instrumented e-scooter. As a result, bicycle tracks having concrete or vegetated curb presented lower clearance distance (≈0.8 m) than those without edge elements (>1 m), with no statistically significant differences found between the interaction with bicycles and e-scooters. Additionally, an online questionnaire was proposed to assess users’ perceived risk during the meeting manoeuvre, concluding that micromobility users feel safer and more comfortable riding on pavements away from parked or moving motorized traffic, and on protected bicycle tracks.

## 1. Introduction

Mobility patterns and lifestyles have changed in recent years in cities worldwide, thanks to the strong rise in modes of travel commonly referred to as micromobility. It includes all transportation modes that allow their users to make a hybrid usage and behave either as a pedestrian or as a vehicle at their convenience or when necessary. Defined as such, microvehicles include all easy-to-carry or easy-to-push vehicles allowing for the augmentation of the pedestrian. They can range from lightest rollers and skis to the heaviest two-wheeled, self-balancing personal transporters. They can be motorized or non-motorized, shared or privately owned [1]. Bicycle riding is the most widespread micromobility transport modes, followed by electric scooters (e-scooters), that can address the first-last mile problem or even be used for door-to-door trips [2].

This mobility change was also influenced by the COVID-19 pandemic. To reduce the risk of becoming infected on public transport, people started to replace public transport with micromobility transport modes, and even the proportion of medium- and long-distance travels by micromobility services increases during the lockdown period [3]. In fact, some studies concluded that there are also indications that micromobility patterns have changed after the pandemic, from complementary modes to full trip solutions [4]. In some cities, since the lockdown, the use of e-scooters has gained significant importance, and has become a strategic means of travel [5]. This has been made possible by the expansion of free-floating (i.e., dockless) e-scooters operators, which have increased its popularity, although COVID put a hold on free-floating e-scooters activities. However, the purchase of scooters by private individuals has risen sharply due to governmental incentives [5]

This great rise of e-scooter users has been accompanied by an increase in crashes involving this micromobility vehicle. However, data availability for e-scooter crashes is very limited. In fact, crashes involving e-scooters do not have dedicated labelling in crash reports for the majority of city agencies [4]. Therefore, most research is based on hospital records and visits to emergency departments [6,7,8,9,10]. Other authors have focused on massive media reports for constructing crash datasets [11], whereas only a few studies rely on police-reported crashes [12]. Although the collision types and severity varied among studies, most of them concluded that the highest percentage of crashes occurs in “sharrows” (a combination of the words “share” and “arrow” referring to roads shared by bikes and cars), followed by sidewalks, with bicycle tracks—striped, buffered or protected—being the safest place for riding.

The model developed by Zhang et al. [13] suggests that e-scooter riders are willing to travel longer distances on segments with bikeways. However, Curl and Fitt [14] highlighted that 90% of users used sidewalks. E-scooter riders’ choices depend on the country, the cities, and their policies. In Spain, e-scooter users are prohibited from riding on sidewalks or pedestrian areas [15], thus most e-scooters ride on bicycle tracks, sharing the facilities with the bicycles.

Existing bicycle tracks shared by e-scooters were originally designated for cyclists. Introducing e-scooters to these facilities undoubtedly causes additional interferences between users, which makes it not only unsafe to e-scooter riders, but also to cyclists and pedestrians. For example, due to physical restrictions (e.g., limited width, roughness, etc), many facilities may not be able to fully support the safe use of e-scooters, which are often equipped with small wheels [16].

E-scooters differ from bikes in terms of dimensions and speed, and these differences can influence not only safety, but also the level-of-service (LOS) of bicycle track. The Bicycle LOS (BLOS) is an important indicator used for bicycle track planning, design, monitoring, prioritisation, and strategy. Without considering the difference in the mobility of e-scooters, e-bikes, and bicycles, the estimates of BLOS could be biased [17].

Consequently, to study and improve safety and LOS of bicycle tracks considering mixed traffic flow, it is necessary to know the users’ behaviour when interacting. These interactions in two-way cycle tracks can be: following, when a faster vehicle reaches a slower one; passing, when, after following, a faster vehicle passes the slower one; and, meeting, when two vehicles traveling in opposing directions cross [18].

Most of the previous studies on either following, passing, or meeting manoeuvres were focused only on bicycles, and based on video recording at fixed locations. Khan and Raksuntorn [19] studied bicycle passing and meeting manoeuvres on a 3 m wide separated bicycle path, comparing speeds of overtaking bicycles at different overtaking states, and analysing clearance distance. The results showed that the average clearance distances during passing and meeting manoeuvres were 1.78 m and 1.94 m, respectively. Mohammed et al. [20] extracted data from video by using computer vision techniques, including longitudinal distance, clearance distance, and speed difference between interacting cyclists, in order to characterize bicycle following and overtaking manoeuvres on cycling paths. They clustered the overtaking cyclists into initiation, merging, and post-overtaking states. The average clearance distance for initiation state was 1.51 m.

Video recording is unobtrusive and allows data collection without influencing cyclists’ behaviour. However, video camera can only be placed in specific locations, thus limiting the generalizability of the findings. Therefore, several studies have used instrumented bicycles to analyse interactions with other vehicles. García et al. [21] observed cyclists’ meeting manoeuvres using an instrumented bicycle, equipped with video cameras, a GPS tracker, laser rangefinders, and speed sensors. They collected data of 336 meeting manoeuvres on six two-way cycle tracks ranging 1.3–2.15 m in width, delimited by different boundary conditions in Valencia (Spain). They found that clearance distance increased with lane width, and decreased as the cycle tracks had lateral obstacles (e.g., parked vehicles or urban furniture). The presence of an obstacle to the wheel height reduced the average clearance distance up to 0.10 m, being 0.20 m in the case of obstacles to the handlebar height. Without obstacles and on wide cycle tracks, the average clearance distance was 0.89 m.

The adaptation of this instrumentation to the e-scooter is complicated due to its size. Garman et al. [22] instrumented an e-scooter to collect performance data related to vehicle dynamics. All data were acquired by a Plex VMU 900 HD Pro equipped with an integrated 50 Hz GPS and 100 Hz Inertial Measurement Unit (IMU) module. Ma et al. [16] developed a system for e-scooter instrumentation integrating a set of sensing devices including GPS, IMU, and Lidar to collect real-time information on geospatial coordinates, vibrations, and surrounding obstacles. All sensors were connected with a Raspberry Pi platform for data acquisition, processing, and storing. However, they studied the interactions between e-scooters and the surrounding environment in urban areas, but not the interactions among bicycle track users.

E-scooter users’ behaviour and their interactions with cyclists during passing and meeting manoeuvres should be a critical issue for bicycle track width selection, for the estimation of BLOS, and for the development of microsimulation models, which can be used to enhance bicycle track planning, traffic modelling, safety assessment, and energy and health modelling. However, previous studies did not analyse the interactions between e-scooters, and between e-scooters and bicycles. To fill this gap, the current study has developed a new sensing system for e-scooter instrumentation. It allows for the study of the behaviour of cyclists and e-scooter users when passing and meeting an e-scooter on urban two-way cycle tracks.

Therefore, this study aims at characterizing the meeting manoeuvre between micromobility users along diverse typologies of two-way bicycle track. For that purpose, an instrumented electric scooter is used to carry out a quasi-naturalistic data collection in the city of Valencia (Spain). Additionally, an online survey is proposed to assess users’ perceived risk associated with meeting manoeuvre.

The underlying hypothesis is that the typology of bicycle track has a great impact on micromobility users’ behaviour. The presence of a physic or vegetated curb is expected to lead to a lower clearance distance between users and, therefore, to a higher perceived risk.

## 2. Materials and Methods

The development of this research is mainly based on four stages: (i) infrastructure analysis; (ii) quasi-naturalistic data collection; (iii) analysis of micromobility users’ behaviour; and (iv) analysis of users’ subjective perception risk.

First, the bike infrastructure network of Valencia city was assessed to identify potential locations to study the interaction between micromobility users. To this regard, cycling demand maps of Valencia, that are publicly available on the website of the Valencia City Council (https://www.valencia.es/cas/movilidad accessed on 6 June 2021), and the typology of bicycle tracks, that was checked on Google Maps, were considered.

Once bicycle track segments to be studied were selected, a quasi-naturalistic data collection was designed and performed by using an instrumented e-scooter. This vehicle allowed estimating clearance distance, determining the type of manoeuvre (meeting or overtaking), and identifying the type of micromobility vehicle involved in each manoeuvre—bicycle or e-scooter.

Although both meeting and overtaking manoeuvres were observed, this study is only focused on meeting manoeuvres, since a low number of overtaking manoeuvres occurred. Meeting manoeuvres were analysed depending on the type of opposite vehicle involved—bike or e-scooter—and the typology of the bicycle track. A descriptive analysis was developed to determine the average and standard deviation of the clearance distance, and, additionally, a statistical analysis was performed to identify whether statistically significant differences exist between the clearance distances regarding the type of vehicle and the type of bicycle track.

Finally, users’ subjective perception risk was assessed through an online questionnaire. The main objective of the questionnaire was to know the users’ perception regarding cycling infrastructure conditions, lane edge conditions, comfort while driving, and interactions with other users (including pedestrians and motorised vehicles).

### 2.1. Bicycle Track Segments

The selection of bicycle tracks was based on the following criteria:
Cycling demand. Traffic volume was high enough to encourage meeting manoeuvres, but not so high to allow overtaking manoeuvres, ensuring free-flow conditions. In this study, bicycle tracks presenting a cycling demand greater than 1000 bikes per day were selected. A maximum value was not established, and those bicycle tracks having very high traffic volume were observed during off-peak hours;Bicycle track typology. More common typologies of bicycle tracks in the city of Valencia were selected (Figure 1). To this regard, sidepath is referred to off-street bikeways that are built as extensions of the sidewalk, with a complete physical separation from cars except at intersections with cross streets. Selected bicycle track typologies are:
(a)Sidepath without physical or vegetated curb;(b)Sidepath with vegetated curb;(c)Protected bicycle track;(d)Sidepath on median next to motorised traffic;(e)Sidepath on median not next to motorised traffic;Bicycle track width. This geometric characteristic was kept constant among the different bicycle track typologies to avoid biased results. After analysing lane width along several two-way bicycle tracks in Valencia, 2 m wide bicycle tracks were selected, since this is the most common bicycle track width;Bicycle track length. A minimum length of 300 m was stablished to ensure free-flow conditions and to encourage both meeting and overtaking manoeuvres. To this regard, data close to intersections—20 m according to AASHTO [23]—were not considered in the analysis.

Table 1 shows a description of selected bicycle tracks, indicating bicycle track typology, location, edge conditions, type of pavement, bicycle track width and length, and cycling demand in June 2021. It should be noted that the configurations of these bicycle tracks are very common not only in Valencia, but also around Spain—e.g., in Madrid and Barcelona—and even cities around Europe and the USA.

### 2.2. Data Collection

Data collection was performed by using an e-scooter equipped with two distance meters (HC-SR04 ultrasonic sensor) controlled through a Raspberry Pi 4—a tiny, dual-display, desktop computer—and a Garmin Virb Elite video camera (Figure 2). Diverse data collection sessions were scheduled from 25 June to 15 July 2021, in the morning between 8:00 h and 9:00 h and in the evening between 17:00 h and 21:00 h. The instrumented vehicle was driven by the same person during all data collection sessions, who travelled in the centre of the directional lane at 15 km/h. In this way, the lateral distance between the instrumented vehicle and the edge of the opposite lane was 1.5 m, since the width of all studied bicycle tracks was 2 m. As a result, a total of 80 km of bicycle tracks were travelled, leading to 25 h of video recording.

### 2.3. Data Reduction

The clearance distance data collected by the ultrasonic sensors were saved in a CSV file through a Python 3.10 script. Specifically, these sensors enabled to measure clearance distance at 10 Hz frequency. These files were opened in Excel to filter and represent the collected information.

Additionally, a video recording for each data collection session was available. These videos were visualized in Garmin Virb Edit, which shows the video recording along with the georeferenced path and the actual speed of the instrumented vehicle.

Both data sources—CSV file and video recording—were synchronized to identify meeting and overtaking manoeuvres (Figure 3). Figure 3a shows clearance distance over time. In this regard, a meeting or overtaking manoeuvre was related to sudden reductions of clearance distance. A clearance distance greater than 150 mm meant that the opposite or overtaking vehicle drove out of the bicycle track during the meeting or overtaking manoeuvre, respectively. Each sudden reduction of clearance distance was checked on the video recordings to identify the following information (Figure 3b): (i) typology of bicycle track; (ii) type of vehicle (bike, e-scooter or other); (iii) type of manoeuvre (meeting or overtaking); (iv) clearance distance; and (v) speed (only available for overtaking manoeuvres).

Table 2 summarizes the number of the meeting and overtaking manoeuvres observed during data collection, considering the diverse typologies of bicycle track and the most common type of vehicles (bikes and e-scooter). As a result, a total of 779 manoeuvres were recorded, of which 93% were meeting manoeuvres. Furthermore, over 70% of micromobility users were traveling by bike.

### 2.4. Survey

An online questionnaire was designed to identify users’ preferences while traveling along the cycling infrastructure from a safety and comfort standpoint. Micromobility users were asked about the risk linked to meeting and overtaking manoeuvres while traveling along different typologies of bicycle track. To this regard, the level of risk was measured through a 5-level Likert scale: (1) no risk; (2) low risk; (3) medium risk; (4) high risk; and (5) very high risk.

A total of 120 micromobility users, aged 18 to 67 years, responded the questionnaire from 17 June 2021 to 20 July 2021 (Figure 4). The questionnaire was launched via e-mail to the academic community and via social networks—Twitter, Facebook, and LinkedIn—to the general audience. As expected, the number of responses decreased with age.

Regarding the main mode of transport of the participants, 35% of them use a private bicycle as their main transport vehicle, 20% use public transport, 17% use the public bicycle system, 14% use their own car, 8% use e-scooters, and only 5% use shared or owned motorcycles (Figure 5). Thus, it can be concluded that six out of ten respondents use a micromobility vehicle—bike or e-scooter—as their main mode of urban transport. In addition, over half of respondents prefer traveling by bike rather than other micromobility vehicles, which is consistent with the trend observed during data collection.

Finally, it should be noted that most of the respondents (62%) travel distances lower than 5 km when using micromobility vehicles, whereas a quarter of them indicated travelling distances between 5 and 10 km. Only about 10% of participants reported travelling distances longer than 10 km.

## 3. Results

Data analysis was focused on exploring the clearance distance between micromobility users and on determining the users’ perceived risk during meeting manoeuvres. Both variables were studied considering bicycle track typology and type of vehicle—bike and e-scooter.

### 3.1. Clearance Distance

First, a descriptive analysis was performed by estimating diverse measures of location and variability (Table 3). Although there are no differences of clearance distance between bikes and e-scooter for a specific typology of bicycle track—no intragroup differences—, it seems that clearance distance is greatly influenced by the typology of bicycle track—intergroup differences. Related to this, bicycle tracks which have a physical or vegetated curb presented a lower clearance distance (≈0.8 m) than those without edge elements (>1 m), for both types of vehicles.

The variability of clearance distance increases as the mean does. In other words, micromobility users seem to feel more freedom on bicycle tracks without edge elements. Indeed, the maximum values of the clearance distance for these typologies of bicycle tracks are higher than 1.5 m, indicating that some oncoming vehicles were outside of the bike infrastructure. Additionally, the variability among cyclists was lower than that linked to users of e-scooters, except for sidepaths with vegetated curb.

Afterwards, a statistical hypothesis test was developed to identify whether statistically significant differences exist between the samples obtained (clearance distances for each type of vehicle per bicycle track typology). For that purpose, the assumption of normality was previously checked considering the Kolmogorov–Smirnov test that establishes as a null hypothesis (H_0_) that the data are normally distributed (Table 4). As a result, most samples were not normally distributed at a 95% confidence level.

Therefore, the non-parametric Kruskal–Wallis test was considered to perform the statistical analysis. It tries to compare the medians of each sample, making all possible combinations to contrast them with each other (Table 5). The intergroup analysis for each type of vehicle indicated that statistically significant differences exist between bicycle tracks with a physical or vegetated curb and those without curbs, verifying one of the main hypotheses of this research.

Although there were no statistically significant differences between the typologies of bicycle track with curb (sidepath with vegetated curb and protected bicycle track), the mean and median values of the clearance distance for protected bicycle tracks were the lowest. This could be due to the proximity of motorised traffic to this type of bicycle track, causing micromobility users to stay away from the lane edge.

Furthermore, no statistically significant differences were identified between the cycle lanes in the median, thus the proximity of motorised traffic to the median did not influence user behaviour.

On the other hand, the intragroup analysis pointed out that cyclists and users of e-scooter behaved similar for a specific typology of bicycle track as the *p*-Values of the non-parametric Kruskal–Wallis test were greater than 0.05 (Table 6).

### 3.2. Perceived Risk

The risk perceived by micromobility users while riding in each typology of bicycle track was analysed by means of an online questionnaire, considering a 5-level Likert scale: (1) no risk; (2) low risk; (3) medium risk; (4) high risk; and (5) very high risk. In this regard, two additional typologies of bicycle track were included to assess the presence of parked motorised vehicles, parked with both parallel and perpendicular parking (Figure 6).

Figure 7 represents, for each typology of bicycle track, the percentage of responses for each value of the Likert scale, placing the average value of the scale (3) at 0%. Thus, the percentages of responses associated with high (4) or very high risk (5), together with half of the percentage of responses associated with medium risk (3), are located on the right-hand side. Similarly, half of the percentage of responses linked to medium risk (3), together with the percentages of responses associated with low (2) or no risk (1), are located on the left-hand side.

The typologies “Sidepath without physical or vegetated curb” and “Sidepath on median” resulted in a large percentage of responses associated with no or low risk (>50%). However, the respondents indicated that the presence of parallel or perpendicular parking in the edge of these typologies of bicycle tracks had a negative impact on safety, significantly raising the perceived risk. In addition, the presence of motorised traffic next to a sidepath on the median led to an increase of the perceived risk. More than one out of four respondents reported a high or very high risk when riding along this type of bicycle track.

Regarding the type of curb, micromobility users felt safer riding on protected bicycle tracks than on a sidepath with vegetated curb. This finding could be due to the lack of maintenance of vegetation that serves to separate cycling infrastructure from pedestrians. However, the clearance distances collected along protected bicycle tracks were lower than those on sidepaths with vegetated curb (Table 3).

Therefore, micromobility users felt safer and more comfortable riding on sidepaths away from parked or moving motorised traffic and on protected bicycle tracks. In particular, the typology of bicycle track associated with the lowest perceived risk was “Sidepath on median”, with a mean value of 2.11. On the contrary, the least safe typology of bicycle track was “Sidepath without physical or vegetated curb, next to perpendicular parking” that resulted in a mean value of perceived risk of 3.71.

## 4. Discussion

The findings of this study are based on the analysis of 728 meeting manoeuvres recorded on seven bicycle track segments. This amount of data is considerably higher than those used in other studies. To this regard, García et al. [21] considered a total of 336 meeting manoeuvres along six bicycle track segments, while Khan and Raksuntorn [19] analysed 100 meeting events on only one bicycle track.

As concluded by Allen et al. [18] and García et al. [21], meeting manoeuvres are the most common events on two-way off-street bicycle tracks. This is consistent with the results of this research since 93% of the recorded manoeuvres were meeting manoeuvres—only 51 passing manoeuvres were reported.

Khan and Raksuntorn [19] concluded that, in a 3 m wide bicycle track lined with trees, the average clearance distance for bicycle meeting events was 1.95 m. According to boundary conditions, this section is similar to the sidepath with vegetated curb section of the current study. For this type of bicycle track, the average clearance distance was 0.837 m and 0.832 m for bikes and e-scooters, respectively, i.e., over one meter lower than the average clearance distance identified by Khan and Raksuntorn [19], similar to the difference between the bicycle track widths of both studies.

Along sidepaths without physical or vegetated curb, the average clearance distance increases to 1.013 m (maximum 2.133 m) for bikes, and 1.054 m (maximum 1.972 m) for e-scooters, reporting that some oncoming vehicles were outside of the bike infrastructure (clearance distances greater than 1.5 m). García et al. [21] also analysed clearance distance by using an instrumented bicycle on this typology of bicycle track, obtaining an average clearance distance of 0.89 m. The difference between the above results could be explained by the difference in bicycle track widths, with the bicycle tracks considered by García et al. [21] being 20 cm narrower (1.8 m wide), and the type of instrumented vehicle.

Moreover, García et al. [21] identified a 0.10 m clearance distance reduction on bicycle tracks with small bushes or curbs and 0.20 m on bicycle tracks next to a line of streetlights. However, the clearance distance reductions observed in this study for those boundary conditions were higher, and varied depending on the obstacle type and on the location of the bicycle track, not only on the height of the obstacle.

Unlike most previous studies focused on bicycle-bicycle interactions, this research analysed the interactions between e-scooter–e-scooter and e-scooter–bicycle. In this regard, the results of this research indicated that no statistically significant differences exist in clearance distance for both type of interactions.

The results of the online survey showed that micromobility users prefer riding on bicycle tracks with no interaction with motorised vehicles—sidepaths on median or without vegetated curbs and protected bicycle track. In this regard, the typology “protected bicycle track” was considered as a low-risky bicycle track despite reporting the lowest clearance distances during the data collection through the instrumented e-scooter.

According to the results of the quasi-naturalistic study and the online survey, some good practices listed in some guidelines for off-road two-lane bicycle tracks have been validated:When designing a bicycle track as an extension of the sidewalk, a separator should be installed to prevent cyclists and e-scooter users from riding in the pedestrian area and vice versa;When designing a bicycle track next to a parking lane, additional space should be left between the parking lane and the bicycle track;On bicycle tracks with bushes or vegetated curb as boundaries, regular maintenance of the vegetation is needed to ensure a proper effective lane width;On bicycle tracks next to motorised traffic, bicycle lane width should be increased to avoid cyclists or e-scooter users falling into the carriageway of motorised vehicle and minimise the aerodynamic impact caused by these vehicles on micromobility users.

## 5. Conclusions

The strong rise of micromobility, especially bicycles and e-scooters, has changed the mobility patterns and lifestyles. These vehicles ride usually along bicycle tracks, sharing the same facilities, with sidewalks being intended for a pedestrian use only. Therefore, research is needed to understand micromobility users’ behaviour, focusing mainly on their interactions (passing and meeting events). Previous studies addressed these interactions, but only focused on bicycles without considering the presence of e-scooters.

This study aimed at characterizing meeting manoeuvres between micromobility users along different types of two-way bicycle tracks physically separated from motorised vehicles. For this purpose, a quasi-naturalistic data collection was developed by using an instrumented e-scooter with two distance meters controlled through a Raspberry Pi 4 and a video camera that allowed estimating lateral clearance distance. As a result, 728 manoeuvres on seven 2 m wide bicycle tracks—grouped in five typologies of bicycle tracks—were identified.

The typologies of bicycle tracks without edge elements—sidepath without physical or vegetated curb, sidepath on median next to motorised traffic, and sidepath on median—encourage cyclists and e-scooter users to ride outside of the bicycle infrastructure, encroaching on the sidewalk. Moreover, they present larger values of both mean clearance distance and standard deviation than the bicycle tracks with physical separation elements. In this regard, the lowest clearance distances were observed on protected bicycle tracks. For all typologies, no statistically significant differences were found when the oncoming vehicle was either a cyclist or an e-scooter user.

Additionally, an online survey was conducted to assess users’ perceived risk associated with the meeting manoeuvre. Two additional typologies of bicycle track were then included to determine the impact of the presence of parked motorised vehicles. The safest bicycle track typologies from the point of view of users were sidepath on median, protected bicycle track, and sidepath without physical or vegetated curb; dramatically raising the users’ perceived risk when parking is allowed in the vicinity of the bicycle track, especially with perpendicular parking. The typologies with highest perceived risk were sidepath with vegetated curb and sidepath on median next to motorised traffic.

Previous research had focused on the interaction only between bicycles. However, although bicycle riding is the most widespread micromobility transport mode, the number of e-scooters’ users has increased worldwide, sharing the infrastructure with bicycles. Therefore, it is important to study the interaction manoeuvres between these micromobility users to improve operation and safety. The present study provides an approach to the behaviour of cyclists and e-scooter users when meeting an e-scooter. Moreover, a new methodology based on an instrumented e-scooter has been developed. This methodology has been applied to two-way bicycle tracks, although it could be applied to the study of micromobility users when riding through other type of configurations.

The present study has been focused on the analysis of meeting manoeuvres between e-scooters, as well as bicycles and e-scooters on five typologies of off-road two-lane bicycle track. Further work is required to include other bicycle track widths and boundary conditions, and to analyse the characterisation of overtaking manoeuvre, not only to e-scooters, but also to bicycles. To do that, an additional data collection is proposed by using the instrumented e-scooter and an instrumented bicycle.

The results of this study could be the basis for the development of guidelines focused on the design of micromobility infrastructures, considering not only bicycles, but also e-scooters. Moreover, this approach to the interactions between micromobility users could be applied to the estimation of BLOS and the development of micromobility microsimulation models. City planners could take these into consideration to enhance bicycle infrastructure planning, traffic modelling, and safety assessment, with the aim of achieving sustainable mobility.

## Figures and Tables

**Figure 1 ijerph-18-12465-f001:**
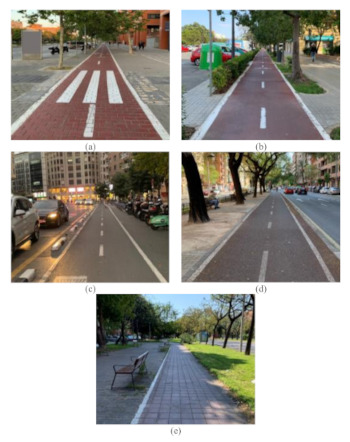
Typologies of bicycle tracks: (**a**) sidepath without physical or vegetated curb, (**b**) sidepath with vegetated curb, (**c**) protected bicycle track, (**d**) sidepath on median next to motorised traffic, and **(e)** sidepath on median.

**Figure 2 ijerph-18-12465-f002:**
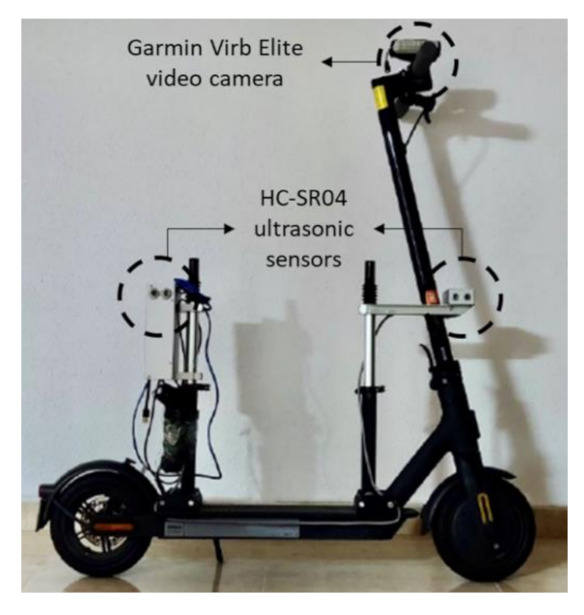
Instrumented e-scooter.

**Figure 3 ijerph-18-12465-f003:**
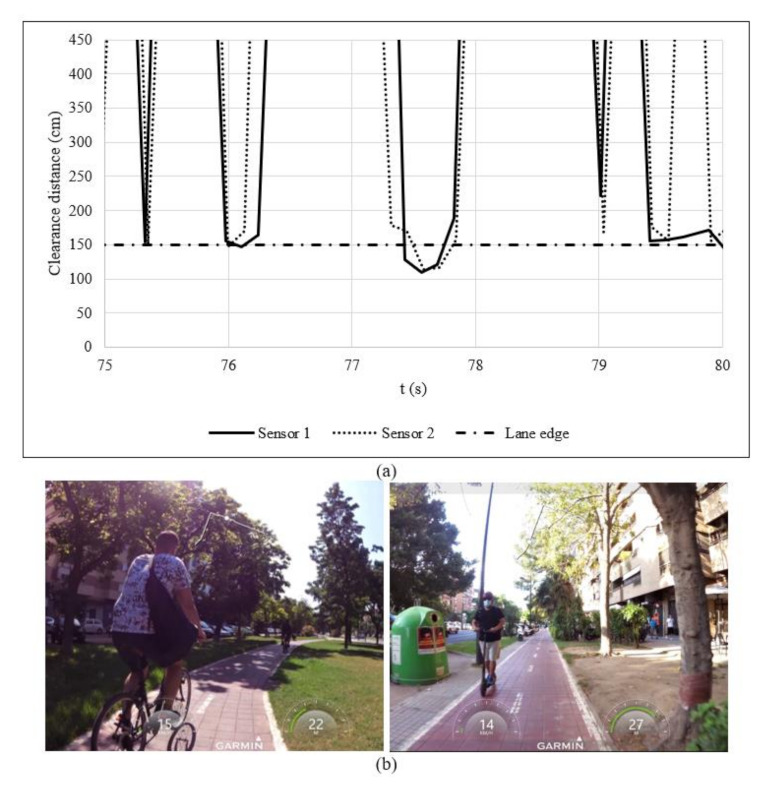
Synchronization of data sources: (**a**) data from ultrasonic sensors and (**b**) video recording.

**Figure 4 ijerph-18-12465-f004:**
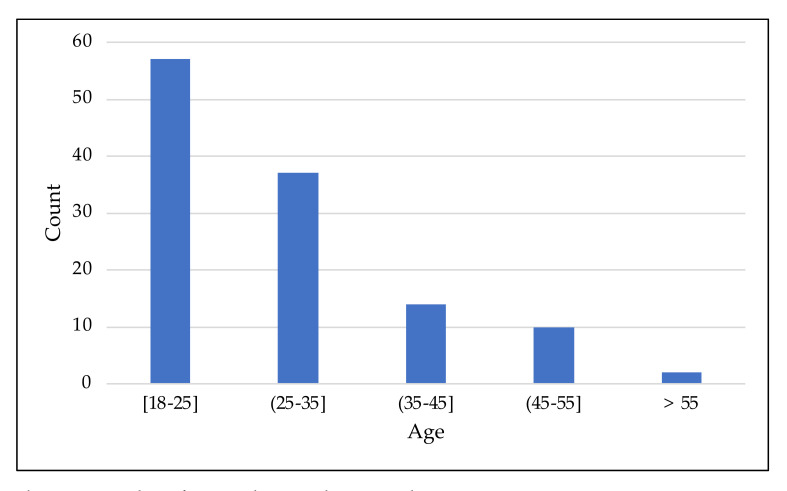
Number of respondents to the survey by age.

**Figure 5 ijerph-18-12465-f005:**
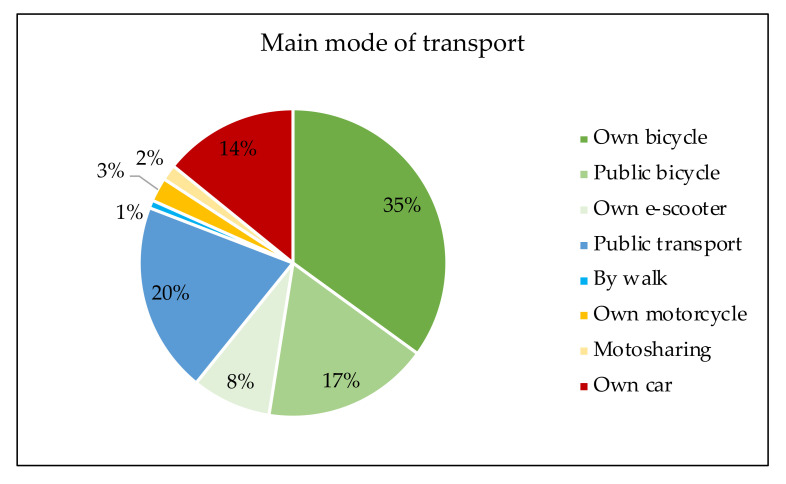
Main mode of transport of the respondents.

**Figure 6 ijerph-18-12465-f006:**
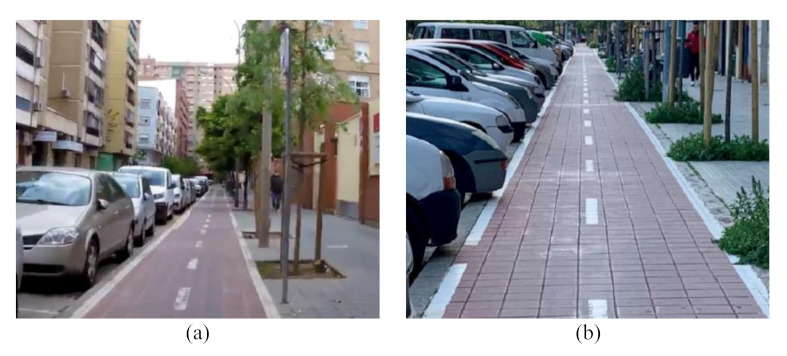
Sidepath without physical or vegetated curb next to parked motorized vehicles: (**a**) parallel parking, and (**b**) perpendicular parking.

**Figure 7 ijerph-18-12465-f007:**
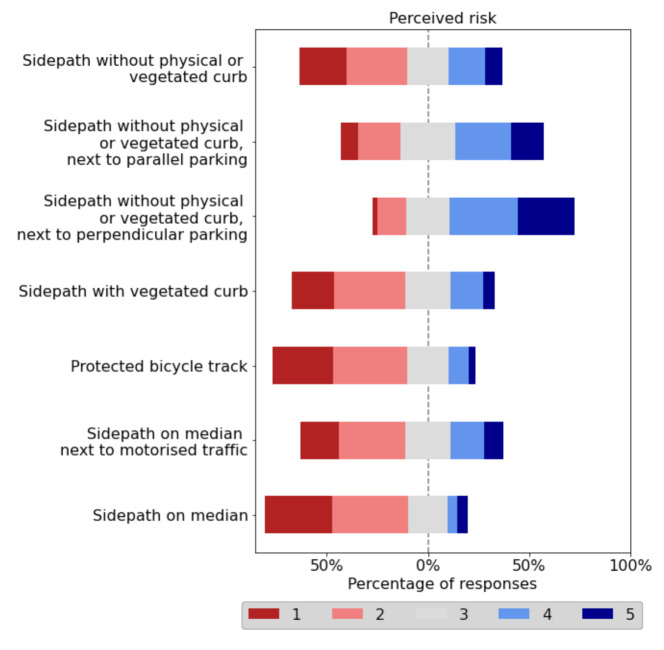
Results of perceived risk during meeting manoeuvres.

**Table 1 ijerph-18-12465-t001:** Selected bicycle tracks.

Id	Typology *	Location	Edge Conditions	Pavement	Width (m)	Length (m)	Cycling Demand (bikes/day)
1	Sidepath ^a^	Naranjos Ave.	-	Cobblestones	2	1500	1230
2	Sidepath ^b^	Naranjos Ave.	Vegetated curb	Concrete	2	1505	1740
3	Sidepath ^b^	Blasco Ibáñez Ave.	Vegetated curb	Concrete	2	480	2064
4	Protected bicycle track ^c^	Colón St.	Discontinuous concrete kerbstone	Asphalt	2	647	4425
5	Protected bicycle track ^c^	Guillem de Castro St.	Discontinuous concrete kerbstone	Asphalt	2	1487	4361
6	Sidepath ^d^	Dr. Manuel Candela St.	On median next to motorised traffic	Asphalt	2	885	1991
7	Sidepath ^e^	Blasco Ibáñez Ave.	On median	Tiles	2	350	2360

* (^a–e^) Indicates the type of bicycle track according to Figure 1.

**Table 2 ijerph-18-12465-t002:** Number of involved vehicles during data collection.

Typology of Bicycle Track	Meeting Manoeuvre		Overtaking Manoeuvre		Total
	Bikes	E-Scooter	Bikes	E-Scooter	
Sidepath without physical or vegetated curb	136	35	4	4	179
Sidepath with vegetated curb	100	32	8	4	144
Protected bicycle track	71	29	3	5	108
Sidepath on median next to motorised traffic	112	60	7	7	186
Sidepath on median	117	36	6	3	162
Total	536	192	28	23	779

**Table 3 ijerph-18-12465-t003:** Statistical summary of clearance distance during meeting manoeuvres.

	Type of Vehicle	Mean (m)	Median (m)	Standard Deviation (m)	Minimum (m)	Maximum (m)
Sidepath without physical or vegetated curb	Bike	1.013	0.933	0.281	0.558	2.133
	e-scooter	1.054	0.982	0.306	0.684	1.972
Sidepath with vegetated curb	Bike	0.837	0.818	0.222	0.287	1.592
	e-scooter	0.832	0.848	0.178	0.405	1.184
Protected bicycle track	Bike	0.761	0.742	0.182	0.380	1.357
	e-scooter	0.770	0.762	0.197	0.382	1.160
Sidepath on median next to motorised traffic	Bike	1.028	0.986	0.265	0.514	1.767
	e-scooter	1.099	1.006	0.341	0.234	1.819
Sidepath on median	Bike	1.073	1.000	0.272	0.615	1.757
	e-scooter	1.079	0.964	0.319	0.534	1.730

**Table 4 ijerph-18-12465-t004:** Results of Kolmogorov–Smirnov test.

Typology of Bicycle Track	*p*-Value *	
	Bike	E-Scooter
Sidepath without physical or vegetated curb	0.0355482	0.0001914
Sidepath with vegetated curb	0.0470537	0.9553620
Protected bicycle track	0.6525390	0.5493860
Sidepath on median next to motorised traffic	0.0150813	0.1207760
Sidepath on median	0.0113474	0.0035132

* If *p*-Value is less than 0.05, H_0_ is rejected at a 95% confidence level.

**Table 5 ijerph-18-12465-t005:** Results of the Kruskal–Wallis test in intergroup analysis.

Typologies of Bicycle Track		Difference	
		Bike	E-Scooter
Sidepath without physical or vegetated curb	Sidepath with vegetated curb	125.276 *	40.929 *
	Protected bicycle track	143.997 *	52.929 *
	Sidepath on median next to motorized traffic	−21.014	−8.038
	Sidepath on median	−39.179	−2.002
Sidepath with vegetated curb	Protected bicycle track	18.721	12.000
	Sidepath on median next to motorized traffic	−146.289 *	−48.967 *
	Sidepath on median	−164.455 *	−42.931 *
Protected bicycle track	Sidepath on median next to motorized traffic	−165.011 *	−60.967 *
	Sidepath on median	−183.176 *	−54.931 *
Sidepath on median next to motorized traffic	Sidepath on median	−18.165	6.0361

* *p*-Value less than 0.05 and H_0_ rejected at a 95% confidence level.

**Table 6 ijerph-18-12465-t006:** Results of the Kruskal–Wallis test in intragroup analysis.

Typology of Bicycle Track	*p*-Value *
Sidepath without physical or vegetated curb	0.3764940
Sidepath with vegetated curb	0.5880840
Protected bicycle track	0.8523590
Sidepath on median next to motorised traffic	0.2173070
Sidepath on median	0.9931360

* If *p*-Value is less than 0.05, H_0_ is rejected at a 95% confidence level.

## Data Availability

Data, models, and code that support the findings of this study are available from the corresponding author upon reasonable request.

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
