# Peer review of "Micromobility Users’ Behaviour and Perceived Risk during Meeting Manoeuvres"

_ijerph, 2021, doi:10.3390/ijerph182312465_

Round 1

Reviewer 1 Report

The manuscript presents an original and very useful theme for road safety.
It is necessary to include the novelty of the research in the introduction and, above all, to consider the heterogeneity of the rules regulating the road use of electric scooters in Europe.

The introduction should also include a preamble on the change in mode choices made by users in the post-pandemic phase, who are increasingly choosing certain modes, including scooters. We therefore recommend reading the following research papers:

1)Li, A., Zhao, P., He, H., & Axhausen, K. W. (2020). Understanding the variations of micro-mobility behavior before and during COVID-19 pandemic period. Arbeitsberichte Verkehrs-und Raumplanung1547.

2)Li, A., Zhao, P., Haitao, H., Mansourian, A., & Axhausen, K. W. (2021). How did micro-mobility change in response to COVID-19 pandemic? A case study based on spatial-temporal-semantic analytics. Computers, Environment and Urban Systems90, 101703.

3)Gkritza, K. (2021). The impact of COVID-19 on user perceptions of public transit, shared mobility/micro-mobility services, and emerging vehicle types [Project].

A description of the possibilities of hiring, sharing or buying these means of transport should also be included, taking into account European incentive policies.
We therefore recommend reading the following research work:

1)Campisi, T., Akgün-Tanbay, N., Nahiduzzaman, M., & Dissanayake, D. (2021, September). Uptake of e-Scooters in Palermo, Italy: Do the Road Users Tend to Rent, Buy or Share?. In International Conference on Computational Science and Its Applications (pp. 669-682). Springer, Cham.

2)Campisi, T., Basbas, S., Skoufas, A., Akgün, N., Ticali, D., & Tesoriere, G. (2020). The impact of COVID-19 pandemic on the resilience of sustainable mobility in Sicily. Sustainability12(21), 8829.

It is considered appropriate to specify the motivation for the choice of the case study (area examined) and to specify whether the work is replicable as a methodology in other contexts.

It is necessary to include a more detailed explanation to accompany Figure 3. 

It is necessary to specify when the questionnaire was administered and how the sample was selected (randomly, experienced users, inhabitants .....).

Probably, for a better reading of figure 5, it is necessary to insert a legend that explains the values from 1 to 5 and to arrange the graphs in a way that the comparison is more immediate.

It is advisable to include the limitations of the research in the final part as well as the possible future steps of investigation, underlining also how these data can be useful as a basis for example to planners or local authorities for the management of the city.

Author Response

Reviwer 1

Comments and Suggestions for Authors

Reviewer (R). The manuscript presents an original and very useful theme for road safety.
It is necessary to include the novelty of the research in the introduction and, above all, to consider the heterogeneity of the rules regulating the road use of electric scooters in Europe.

Authors (A). First of all, the authors would like to thank the reviewer for the comments and suggestions so as to enhance the quality and understanding of the manuscript.

R - The introduction should also include a preamble on the change in mode choices made by users in the post-pandemic phase, who are increasingly choosing certain modes, including scooters. We therefore recommend reading the following research papers:

1)Li, A., Zhao, P., He, H., & Axhausen, K. W. (2020). Understanding the variations of micro-mobility behavior before and during COVID-19 pandemic period. Arbeitsberichte Verkehrs-und Raumplanung1547.

2)Li, A., Zhao, P., Haitao, H., Mansourian, A., & Axhausen, K. W. (2021). How did micro-mobility change in response to COVID-19 pandemic? A case study based on spatial-temporal-semantic analytics. Computers, Environment and Urban Systems90, 101703.

3)Gkritza, K. (2021). The impact of COVID-19 on user perceptions of public transit, shared mobility/micro-mobility services, and emerging vehicle types [Project].

A - The authors have revised the content of these documents and the following paragraph has been added to the manuscript concerning the influence of the pandemic of COVID-19 on micromobility patterns.

“This mobility change was also influenced by the pandemic of COVID-19. To reduce the risk of getting infected in public transport, people started to replace public transport with micromobility transport modes, and even the proportion of medium- and long-distance travels by micromobility services increases during the lockdown period [3]. In fact, some studies concluded that there are also indications that micromobility patterns have changed after the pandemic, from complementary modes to full trip solutions [4]. In some cities, since the lockdown, the use of e-scooters has gained significant importance and has become a strategic mean of travel [5]. This has been made possible by the expansion of free-floating (i.e., dockless) e-scooters operators, which increased its popularity, although COVID put on hold free-floating e-scooters activities. However, the purchase of scooters by private individuals has risen sharply due to governmental incentives [5].”

R - A description of the possibilities of hiring, sharing or buying these means of transport should also be included, taking into account European incentive policies.
We therefore recommend reading the following research work:

1)Campisi, T., Akgün-Tanbay, N., Nahiduzzaman, M., & Dissanayake, D. (2021, September). Uptake of e-Scooters in Palermo, Italy: Do the Road Users Tend to Rent, Buy or Share?. In International Conference on Computational Science and Its Applications (pp. 669-682). Springer, Cham.

2)Campisi, T., Basbas, S., Skoufas, A., Akgün, N., Ticali, D., & Tesoriere, G. (2020). The impact of COVID-19 pandemic on the resilience of sustainable mobility in Sicily. Sustainability12(21), 8829.

A - The authors have revised both manuscripts and information related to the possibilities of hiring, sharing, or buying micromobility vehicles has been added in Introduction section as follows.

“This mobility change was also influenced by the pandemic of COVID-19. To reduce the risk of getting infected in public transport, people started to replace public transport with micromobility transport modes, and even the proportion of medium- and long-distance travels by micromobility services increases during the lockdown period [3]. In fact, some studies concluded that there are also indications that micromobility patterns have changed after the pandemic, from complementary modes to full trip solutions [4]. In some cities, since the lockdown, the use of e-scooters has gained significant importance and has become a strategic mean of travel [5]. This has been made possible by the expansion of free-floating (i.e., dockless) e-scooters operators, which increased its popularity, although COVID put on hold free-floating e-scooters activities. However, the purchase of scooters by private individuals has risen sharply due to governmental incentives [5].”

R - It is considered appropriate to specify the motivation for the choice of the case study (area examined) and to specify whether the work is replicable as a methodology in other contexts.

A - Valencia is a city that encourages the use of micromobility thanks to its orography (flat terrain) and space distribution. In addition, the configurations of the different bicycle tracks are very similar to those located in cities like Madrid or Barcelona, and even in cities around Europe and the US.

Regarding this, the following information has been added in the manuscript:

“Table 1 shows a description of selected bicycle tracks, indicating bicycle track typology, location, edge conditions, type of pavement, bicycle track width and length, and cycling demand in June 2021. It should be noted that the configurations of these bicycle tracks are very common not only in Valencia, but also around Spain –e.g., Madrid and Barcelona– and even cities around Europe and the US.”

Moreover, the contribution of the innovative data collection developed in this study has been highlighted in Conclusions section:

“Besides, a new methodology based on an instrumented e-scooter has been developed. This methodology has been applied to two-way bicycle tracks, but it could be applied to the study of micromobility users when riding through other type of configurations.”

R - It is necessary to include a more detailed explanation to accompany Figure 3.

A - More detailed information about data reduction has been added regarding Figure 3 by explaining Figure 3a and Figure 3b.

“Both data sources –CSV file and video recording– were synchronized to identify meeting and overtaking manoeuvres (Figure 3). Figure 3a shows clearance distance over time. To this regard, a meeting or overtaking manoeuvre was related to sudden reductions of clearance distance. A clearance distance greater than 150 mm means that the opposite or overtaking vehicle drove out the bicycle track during the meeting or over-taking manoeuvre, respectively. Each sudden reduction of clearance distance was checked on the video recordings to identify the following information (Figure 3b): (i) typology of bicycle track; (ii) type of vehicle (bike, e-scooter or other); (iii) type of ma-noeuvre (meeting or overtaking); (iv) clearance distance; (v) speed (only available for overtaking manoeuvres).”

R - It is necessary to specify when the questionnaire was administered and how the sample was selected (randomly, experienced users, inhabitants .....).

A - Additional data about the survey was included in the manuscript. To this regard, the dates that the survey was active, the distribution of respondents by age (Figure 4), the main mode of transport (Figure 5), and the usual distances travelled are now in 2.4 Survey section.

“A total of 120 micromobility users, aged 18 to 67 years, responded the questionnaire from 17 June 2021 to 20 July 2021 (Figure 4). The questionnaire was launched via e-mail to the academic community and via social networks –Twitter, Facebook, and LinkedIn– to the general audience. As expected, the number of responses decreased with the age.

Regarding the main mode of transport of the participants, 35% of them use the private bicycle as the main transport vehicle, 20% use public transport, 17% use the public bicycle system, 14% use their own car, 8% use e-scooters, and only 5% use shared or owned motorcycles. Thus, it can be concluded that six out of ten respondents use a micromobility vehicle –bike or e-scooter– as their main mode of urban transport. In addition, over a half of respondents prefers traveling by bike rather than other micromobility vehicles, which is consistent with the trend observed during data collection.

Finally, it should be noted that most of respondents (62%) travel distances lower than 5 km when using micromobility vehicles, whereas a quarter of them indicated to travel distances between 5 and 10 km. Only about 10% of participants reported travelling distances longer than 10 km.”

R - Probably, for a better reading of figure 5, it is necessary to insert a legend that explains the values from 1 to 5 and to arrange the graphs in a way that the comparison is more immediate.

A - Although the meaning of the values included in Figure 5 were explained in 2.4 Survey section, it has been added again at the beginning of 3.2 Perceived risk section. In addition, Figure 5 (now Figure 7) has been edited to enhance its understanding.

R - It is advisable to include the limitations of the research in the final part as well as the possible future steps of investigation, underlining also how these data can be useful as a basis for example to planners or local authorities for the management of the city.

A - The contribution, limitations, and further research included in Conclusions section have been rewritten as follows, including more detailed information:

“Previous research had focused on the interaction only between bicycles. However, although bicycle riding is the most widespread micromobility transport mode, the number of e-scooters’ users has increased worldwide sharing the infrastructure with bicycles. Therefore, it is important to study the interaction manoeuvres between these micromobility users to improve operation and safety. The present study provides an approach to the behaviour of cyclists and e-scooter users when meeting an e-scooter. Besides, a new methodology based on an instrumented e-scooter has been developed. This methodology has been applied to two-way bicycle tracks, but it could be applied to the study of micromobility users when riding through other type of configurations.

The present study has been focused on the analysis of meeting manoeuvres between e-scooters and bicycles and e-scooters on five typologies of off-road two-lane bicycle track. Further work is required to include other bicycle track widths and boundary conditions and to analyse the characterisation of overtaking manoeuvre, not only to e-scooters but also to bicycles. To do that, an additional data collection is proposed by using the instrumented e-scooter and an instrumented bicycle.

The results of this study could be the basis for the development of guidelines focused on the design of micromobility infrastructures, considering not only bicycles but also e-scooters. Besides, this approach to the interactions between micromobility users could be applied to the estimation of BLOS and the development of micromobility microsimulation models. City planners could take these into consideration to enhance bicycle infrastructure planning, traffic modelling, and safety assessment, with the aim of achieving sustainable mobility.”

Reviewer 2 Report

Dear authors,

I find your research very interesting with much potential for further investigation.

Please take into account some minor comments and suggestions:

  • Please expand the presentation of your questionnaire survey, especially in terms of sampling method and respondents' characteristics (e.g. distribution of sample by age, gender but also mode and trip choices for daily travel). This would apply also to the presentation of survey results.
  • On line 258, you mention: "The variability of clearance distance increases with the mean does." Please check if there is a syntax error in the sentence.
  • Your suggestions in the Section: "Discussion" may be validated by your work but they are really general design principles that would apply, according to standards and specific conditions, in many cases. This does not do justice to your work, that can be used by in many ways by planners, policy makers and travel behaviour analysts. I would suggest to briefly mention or exclude these suggestions.
  • In return, I would suggest to enrich your conclusions in two ways: Mention the added value of your approach to the above and/or other members of the scientific and civic community. Suggest some directions for follow-up research.

Thank you in advance for taking the above comments into consideration.

Author Response

Reviewer 2

Comments and Suggestions for Authors

Reviewer (R) - Dear authors,

I find your research very interesting with much potential for further investigation.

Authors (A) - First of all, the authors would like to thank the reviewer for the comments and suggestions so as to enhance the quality and understanding of the manuscript.

R - Please take into account some minor comments and suggestions:

Please expand the presentation of your questionnaire survey, especially in terms of sampling method and respondents' characteristics (e.g. distribution of sample by age, gender but also mode and trip choices for daily travel). This would apply also to the presentation of survey results.

A - Additional data about the survey was included in the manuscript. To this regard, the dates that the survey was active, the distribution of respondents by age (Figure 4), the main mode of transport (Figure 5), and the usual distances travelled are now in 2.4 Survey section.

“A total of 120 micromobility users, aged 18 to 67 years, responded the questionnaire from 17 June 2021 to 20 July 2021 (Figure 4). The questionnaire was launched via e-mail to the academic community and via social networks –Twitter, Facebook, and LinkedIn– to the general audience. As expected, the number of responses decreased with the age.

Regarding the main mode of transport of the participants, 35% of them use the private bicycle as the main transport vehicle, 20% use public transport, 17% use the public bicycle system, 14% use their own car, 8% use e-scooters, and only 5% use shared or owned motorcycles. Thus, it can be concluded that six out of ten respondents use a micromobility vehicle –bike or e-scooter– as their main mode of urban transport. In addition, over a half of respondents prefers traveling by bike rather than other micromobility vehicles, which is consistent with the trend observed during data collection.

Finally, it should be noted that most of respondents (62%) travel distances lower than 5 km when using micromobility vehicles, whereas a quarter of them indicated to travel distances between 5 and 10 km. Only about 10% of participants reported travelling distances longer than 10 km.”

R - On line 258, you mention: "The variability of clearance distance increases with the mean does." Please check if there is a syntax error in the sentence.

A - This sentence has been rewritten as follows:

“The variability of clearance distance increases as the mean does.”

R - Your suggestions in the Section: "Discussion" may be validated by your work but they are really general design principles that would apply, according to standards and specific conditions, in many cases. This does not do justice to your work, that can be used by in many ways by planners, policy makers and travel behaviour analysts. I would suggest to briefly mention or exclude these suggestions.

A - The authors agree with the reviewer in this issue. Thus, the introduction to these good practices has been rewritten as follows:

“According to the results of the quasi-naturalistic study and the online survey, some good practices listed in some guidelines for off-road two-lane bicycle tracks have been validated:”

R - In return, I would suggest to enrich your conclusions in two ways: Mention the added value of your approach to the above and/or other members of the scientific and civic community. Suggest some directions for follow-up research.

A - The contribution of the study and further research have been clarified and extended in Conclusions section:

“Previous research had focused on the interaction only between bicycles. However, although bicycle riding is the most widespread micromobility transport mode, the number of e-scooters’ users has increased worldwide sharing the infrastructure with bicycles. Therefore, it is important to study the interaction manoeuvres between these micromobility users to improve operation and safety. The present study provides an approach to the behaviour of cyclists and e-scooter users when meeting an e-scooter. Besides, a new methodology based on an instrumented e-scooter has been developed. This methodology has been applied to two-way bicycle tracks, but it could be applied to the study of micromobility users when riding through other type of configurations.

The present study has been focused on the analysis of meeting manoeuvres between e-scooters and bicycles and e-scooters on five typologies of off-road two-lane bicycle track. Further work is required to include other bicycle track widths and boundary conditions and to analyse the characterisation of overtaking manoeuvre, not only to e-scooters but also to bicycles. To do that, an additional data collection is proposed by using the instrumented e-scooter and an instrumented bicycle.

The results of this study could be the basis for the development of guidelines focused on the design of micromobility infrastructures, considering not only bicycles but also e-scooters. Besides, this approach to the interactions between micromobility users could be applied to the estimation of BLOS and the development of micromobility microsimulation models. City planners could take these into consideration to enhance bicycle infrastructure planning, traffic modelling, and safety assessment, with the aim of achieving sustainable mobility.”

R - Thank you in advance for taking the above comments into consideration.

Round 2

Reviewer 1 Report

The manuscript still has some typos and grammatical errors 
Once this is corrected, the paper will be eligible for publication.